# Point Cloud Dataset Distillation

**Deyu Bo** [1]    **Xinchao Wang** [1]

## Abstract

This study introduces dataset distillation (DD) tailored for 3D data, particularly point clouds. DD aims to substitute large-scale real datasets with a small set of synthetic samples while preserving model performance. Existing methods mainly focus on structured data such as images. However, adapting DD for unstructured point clouds poses challenges due to their diverse orientations and resolutions in 3D space. To address these challenges, we theoretically demonstrate the importance of matching rotation-invariant features between real and synthetic data for 3D distillation. We further propose a plug-and-play point cloud rotator to align the point cloud to a canonical orientation, facilitating the learning of rotation-invariant features by all point cloud models. Furthermore, instead of optimizing fixed-size synthetic data directly, we devise a point-wise generator to produce point clouds at various resolutions based on the sampled noise amount. Compared to conventional DD methods, the proposed approach, termed DD3D, enables efficient training on low-resolution point clouds while generating high-resolution data for evaluation, thereby significantly reducing memory requirements and enhancing model scalability. Extensive experiments validate the effectiveness of DD3D in shape classification and part segmentation tasks across diverse scenarios, such as cross-architecture and cross-resolution settings.

## 1. Introduction

Dataset distillation (DD) (Wang et al., 2018) aims to distill the knowledge of a large-scale dataset into a few synthetic samples, where the models trained on the real and synthetic data will have comparable performance. By doing so, DD

significantly reduces the computational cost of training neural networks from scratch. Due to its remarkable efficiency and effectiveness, DD has been used in a variety of domains, such as image (Zhao et al., 2021; Zhao & Bilen, 2023; Cazenavette et al., 2022), video (Wang et al., 2024), graph (Jin et al., 2022; Liu et al., 2024), etc.

Despite great progress, existing DD methods have primarily been applied to structured 1D and 2D data, while the distillation of unstructured 3D data, *e.g.*, point cloud, remains largely unexplored. Point cloud data exists in large quantities in machine vision. For example, MVPNet (Yu et al., 2023) scans more than 87K point clouds from real-world videos, and Objaverse-XL (Deitke et al., 2023) provides more than 10M high-quality 3D assets. Training on such datasets from scratch is time- and resource-intensive, highlighting the need for more efficient alternatives.

However, extending DD to 3D point clouds presents unique challenges. First, point clouds with different orientations represent the same semantic information, *e.g.*, shapes. However, existing DD methods do not take the symmetry of data into account, which cannot handle the randomly rotated data and result in sub-optimal performance. As shown in Figure 1(a), directly applying DD to the point clouds with different orientations cannot obtain meaningful synthetic data. Second, point clouds have flexible resolutions, *i.e.*, the number of points, depending on specific models and applications. Generally, a larger resolution encodes more fine-grain information but also increases the computational costs (Huang et al., 2024; Qiu et al., 2021). Existing DD methods initialize the synthetic data as a fixed-size tensor, which cannot be customized for different point cloud models. Moreover, the memory budget for fixed-size tensors will increase rapidly when dealing with dense-resolution scenes, *e.g.*, segmentation (Chang et al., 2015).

Once the weaknesses of existing methods are identified, it is natural to ask: *How can we build a distillation framework that overcomes the orientation and resolution issues of 3D point clouds?* To answer this question, we first theoretically prove that random rotations weaken the principle components of real data, thereby degenerating the distillation performance. Based on this discovery, we propose **DD3D**, the first DD framework for 3D point clouds, illustrated in Figure 1(b). Specifically, DD3D first uses a rotator to convert the point cloud into a canonical orientation by

---

[1]National University of Singapore. Correspondence to: Xinchao Wang <xinchao@nus.edu.sg>.

*Proceedings of the 42$^{nd}$ International Conference on Machine Learning*, Vancouver, Canada. PMLR 267, 2025. Copyright 2025 by the author(s).

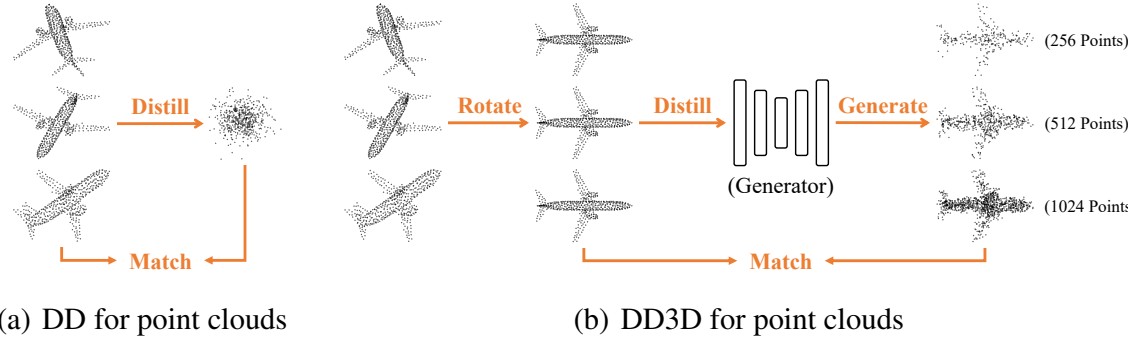

Figure 1: Differences between vanilla DD and DD3D when distilling 3D point clouds.

learning a rotation-equivariant projection matrix to offset random rotation. Then, the knowledge of rotation-invariant data is distilled into a point-wise generator to predict the point coordinates from noise, where the resolution is based on the number of sampled noises. Finally, the rotator and generator are jointly optimized by minimizing the gradient differences between the real and synthetic data.

The contributions are summarized as follows. (1) We propose the first 3D distillation framework, DD3D, which can eliminate the influence of random rotations and synthesize point clouds at arbitrary resolutions. (2) We theoretically prove that matching the rotation-invariant features can preserve the principal components of real data and prevent data degeneration. (3) DD3D can be trained with low-resolution point clouds and generates high-resolution data for evaluation, significantly reducing memory usage and enhancing model scalability. (4) Extensive experiments on shape classification and part segmentation tasks validate the effectiveness of DD3D over baselines.

## 2. Related Work

**Dataset Distillation.** Research on DD can be roughly divided into two directions. The first is to explore advanced matching objectives to improve the distillation performance. For example, performance matching (Wang et al., 2018), gradient matching (Zhao et al., 2021; Zhao & Bilen, 2021), distribution matching (Zhao & Bilen, 2023; Wang et al., 2022), trajectory matching (Cazenavette et al., 2022; Guo et al., 2024; Du et al., 2023) and feature regression (Zhou et al., 2022; Loo et al., 2022; Nguyen et al., 2021). On the other hand, some methods innovate efficient data parameterizations to avoid directly optimizing the synthetic data. For example, neural networks (Liu et al., 2022), spectral representation (Shin et al., 2023), linear transformation (Deng & Russakovsky, 2022), and up-sampling (Kim et al., 2022). Among them, a special parameterization technique is to distill the knowledge into a generative model (Zhao & Bilen, 2022; Wang et al., 2023; Zhang et al., 2023; Cazenavette

et al., 2023; Zhang et al., 2024a), which can generate diverse synthetic data with unlimited samples. Although valid, these methods rely on the prior knowledge of generative models pre-trained on large-scale datasets, which is not feasible for point clouds. A recent work[1] also applies GM to point cloud data. However, neither of them considers the orientation and resolution issues. For a more detailed introduction to DD, please refer to the recent surveys (Yu et al., 2024b; Lei & Tao, 2024; Geng et al., 2023; Sachdeva & McAuley, 2023).

**Point Cloud Analysis.** Deep learning on point clouds plays a vital role in 3D data analysis (Guo et al., 2021b). Traditional methods can be classified into three categories: Point-based methods, *e.g.*, PointNet (Qi et al., 2017a) and PointNet++ (Qi et al., 2017b), convolution-based methods, *e.g.*, PointCNN (Li et al., 2018) and PointConv (Wu et al., 2019), and relation-based methods, *e.g.*, DGCNN (Wang et al., 2019) and Point Transformer (Guo et al., 2021a). However, these methods are rotation-sensitive and cannot handle point clouds with different orientations. Some advanced methods are designed to learn rotation-equivariant or invariant features, such as vector neuron (Deng et al., 2021), spherical harmonic (Poulenard et al., 2019), tensor field (Thomas et al., 2018), and graph features (Kim et al., 2020; Zhao et al., 2019). However, these methods introduce additional operators and cannot be applied to rotation-sensitive methods. Another way is to project point clouds into the same orientation. For example, principal component analysis (PCA) leverages the eigenvectors of the covariance matrix to transform point clouds into the direction with maximum variance. But this approach suffers from the sign-ambiguity issue (Xiao et al., 2020; Yu et al., 2020; Li et al., 2021).

## 3. Preliminary

**Task Formulation.** Suppose that $\mathcal{T} = \{(\mathcal{C}_i, y_i)\}_{i=1}^{|\mathcal{T}|}$ is a large-scale training dataset, where $\mathcal{C}_i$ is a point cloud with label $y_i$ for the shape classification task. Each point

---

[1] https://github.com/kghandour/dd3d

cloud has $n$ points, represented as $\mathcal{C} = \{P, V\}$, where $P \in \mathbb{R}^{n \times 3}$ represents the 3D coordinates of points and $V \in \mathbb{R}^{n \times v}$ indicates the part to which the point belongs in segmentation task and $v$ is the number of parts. The goal of DD3D is to synthesize a much smaller point cloud dataset $\mathcal{S} = \{(\mathcal{C}_j, y_j)\}_{j=1}^{|\mathcal{S}|}$, where $|\mathcal{S}| \ll |\mathcal{T}|$, such that a classification or segmentation model $f_\theta$ trained on $\mathcal{T}$ and $\mathcal{S}$ will have comparable performance. Other tasks, such as detection, are left for future studies.

**Dataset Distillation.** In order to effectively optimize the synthetic data, existing DD methods adopt a bi-level optimization paradigm, which can be formulated as:

$$\min_{\mathcal{S}} \mathcal{L}_{DD}\left(f_{\theta*}(\mathcal{S}), f_{\theta*}(\mathcal{T})\right) \quad (1)$$

$$\text{s.t.} \quad \theta* = \arg\min_{\theta} \mathcal{L}_{cls}(f_\theta(\mathcal{S}), Y^{\mathcal{S}}), \quad (2)$$

where the inner loop updates the model $f_\theta$ on the synthetic data, and the outer loop optimizes the synthetic data. In particular, $\mathcal{L}_{DD}$ is a metric that measures the distance between real and synthetic data. For example, gradient matching (Zhao et al., 2021) minimizes the gradient differences.

**Dataset Distillation with Rotations.** Before detailing the proposed method, we first give a general analysis of how rotations affect the performance of DD. Let $X_{\mathcal{S}} \in \mathbb{R}^{|\mathcal{S}| \times d}$, $X_{\mathcal{T}} \in \mathbb{R}^{|\mathcal{T}| \times d}$ denote the representations learned by $f_\theta$ on the synthetic data and real training data, respectively, and $d$ is the hidden dimension.

**Theorem 3.1.** *Assume the classifier is a linear layer $W$ and $\mathcal{L}_{cls}$ can be simplified to the mean-squared error $\|XW - Y\|_F^2$. The objective of gradient matching is equal to variance preserving:*

$$\min_{\mathcal{S}} \mathcal{L}_{GM} = \min_{\mathcal{S}} \mathcal{D}\left(\nabla_W \mathcal{L}_{cls}^{\mathcal{S}}, \nabla_W \mathcal{L}_{cls}^{\mathcal{T}}\right) \quad (3)$$

$$\Rightarrow \quad \min_{\mathcal{S}} \left\| X_{\mathcal{S}}^\top X_{\mathcal{S}} - X_{\mathcal{T}}^\top X_{\mathcal{T}} \right\|_F^2, \quad (4)$$

*where $\mathcal{D}$ is a distance metric and $\nabla_W$ is the gradient with respect to $W$.*

Theorem 3.1 reveals that synthetic data preserves the variance information of real data. We then analyze how random rotations affect the variance of real data. Without loss of generality, we assume that $f_\theta$ is rotation-equivariant, *i.e.*, $f_\theta(PR) = f_\theta(P)R$, where $R \in \mathrm{SO}(d)$ is a random rotation matrix.

**Theorem 3.2.** *Assume $X_{\mathcal{T}}$ follows a d-dimensional multivariate Gaussian distribution $\mathcal{N}(\boldsymbol{\mu}, \Sigma)$. Let $X_{\mathcal{T}}'$ be the rotated representations of $X_{\mathcal{T}}$ such that:*

$$\lambda_{max}\left(\mathbb{E}\left[X_{\mathcal{T}}'^\top X_{\mathcal{T}}'\right]\right) \leq \lambda_{max}\left(\mathbb{E}\left[X_{\mathcal{T}}^\top X_{\mathcal{T}}\right]\right) \quad (5)$$

$$\Rightarrow \quad \sigma_{max}\left(\mathbb{E}\left[X_{\mathcal{T}}'\right]\right) \leq \sigma_{max}\left(\mathbb{E}\left[X_{\mathcal{T}}\right]\right), \quad (6)$$

*where $\lambda_{max}$ and $\sigma_{max}$ are the maximum eigenvalues and singular values, respectively.*

Theorem 3.2 states that random rotations reduce the maximum singular value of the data representations, implying that the principle component of $X_{\mathcal{T}}$ is weakened. In this case, the synthetic data cannot effectively capture the distribution of the real data, degenerating model performance. All proofs can be seen in Appendix A.

# 4. The Proposed Method

## 4.1. Plug-and-Play Point Cloud Rotator

Our analysis highlights the importance of learning rotation-invariant representations for effective point cloud distillation. However, many existing point cloud models lack this capability. To address this limitation, we introduce a plug-and-play point cloud rotator that transforms point clouds into a consistent canonical view. This transformation ensures that all models can learn rotation-invariant representations, enhancing their generalization and performance.

**Rotation-equivariance.** We leverage the orthogonality of the rotation matrix to eliminate its influence, *i.e.*, $RR^\top = I$, where PCA is a typical method:

$$\frac{1}{n} \sum \left(PR - \overline{P}R\right)^\top \left(PR - \overline{P}R\right) = R^\top U \Lambda U^\top R, \quad (7)$$

where $\overline{P}$ is the center of $P$ and $U$ represents the eigenvectors of the covariance matrix. Importantly, the projection $R^\top U$ maintains equivariance with respect to coordinate rotations, ensuring $(PR)(R^\top U) = PU$ remains invariant. However, eigenvectors suffer from sign ambiguity, implying that both $u_i$ and $-u_i$ are valid solutions. As a result, the canonical view $PU$ is not unique and has 8 ambiguities in 3D space (Xiao et al., 2020; Yu et al., 2020), *i.e.*, $PUQ = P[\pm u_1, \pm u_2, \pm u_3]$, where $\{Q \in \mathbb{R}^{3 \times 3} | Q_{ii} = \{1, -1\}, Q_{ij} = 0, \forall i \neq j\}$.

**Sign-invariant.** Our proposed rotator $r : \mathbb{R}^{n \times 3} \rightarrow \mathbb{R}^{n \times 3}$ is designed to enhance PCA by addressing the sign ambiguity issue. To achieve this, the rotator learns a sign-equivariant reflection matrix $\overline{Q}$ for each point cloud. This ensures that the transformed representation satisfies $PUQ \cdot \overline{Q} = PU$, making it sign-invariant and improving the robustness of rotation-invariant learning. Specifically, the rotator first lifts the scalar coordinates to the vector representations:

$$H = [\sin(\pm PU), \sin(\pm 2PU) \cdots \sin(\pm tPU)]^\top$$
$$= [\sin(PU), \sin(2PU) \cdots \sin(tPU)]^\top Q, \quad (8)$$

where $\sin(\cdot)$ is the sine function and $t$ is the period of Fourier features. An average pooling is then applied on $H$ to learn the representations of the whole point cloud. Finally, a learnable vector $w \in \mathbb{R}^t$ is used to decode the reflection matrix. The overall architecture of the rotation is formulated as follows:

$$r(P) = PUQ \cdot \overline{Q} = PUQ \cdot \text{Sign}(w \cdot \text{Pool}(HQ)), \quad (9)$$

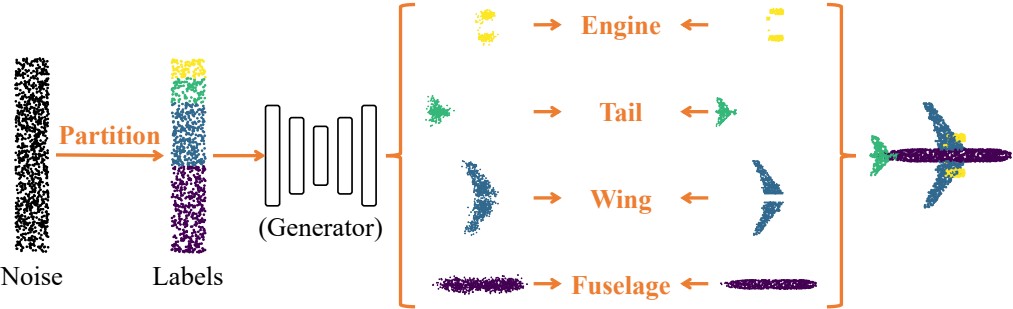

Figure 2: DD3D for part segmentation task. Each noise is first pre-partitioned into different parts according to its value, *e.g.*, the noise within (0, 0.45) is marked as fuselage. Then the generator maps the noise into different parts for gradient matching.

where Sign means the signs of a matrix. The reflection matrix $\overline{Q}$ has the same signs as $Q$ because the sinusoidal features, pooling function, and linear decoder preserve the sign information of $HQ$, which can solve the sign ambiguity and learn sign-invariant representations.

**Alternative Approaches.** Several methods (Zhang et al., 2024b; Yu et al., 2024a; Melnyk et al., 2024; Li et al., 2022; Xu et al., 2021) have been proposed for learning rotation-invariant representations, such as vector neurons (Deng et al., 2021) and graph-based features (Kim et al., 2020). However, these approaches modify the original point coordinates, making it difficult to integrate with existing models. Another line of work addresses the sign ambiguity issue using pooling (Yu et al., 2020) and attention mechanisms (Xiao et al., 2020; Li et al., 2021). While effective, these methods are computationally expensive, as they require evaluating representations across all possible ambiguous views.

### 4.2. Point-wise Generator

Beyond rotation alignment, point cloud distillation must also account for the variations in resolution. Unlike images, point clouds do not have a fixed structure, making traditional DD methods, which directly optimize fixed-size tensors, unsuitable. To solve this issue, a promising solution is to parameterize data with implicit neural representation (INR), which has been widely used to generate data at arbitrary resolutions (Sitzmann et al., 2020; Park et al., 2019; Chen et al., 2021; Singh et al., 2023).

**Point Denoising.** Our solution is to use INR as a point-wise generator $g : \mathbb{R} \to \mathbb{R}^3$, which takes a random noise as input and predicts its corresponding 3D coordinates. Therefore, the number of points is the same as the sampling noise, which enables us to achieve low-resolution training and high-resolution evaluation, thus significantly reducing the computational costs and memory budget. See Section 5.6 for more details. For implementation, we choose SIREN (Sitzmann et al., 2020) as generator, which is formulated as

$$g = [\Phi_1 \circ \Phi_2 \circ \cdots \circ \Phi_L] W_P, \ \Phi_i = \sin(z_i w_i + b_i), \quad (10)$$

where $L$ is the number of layers, $\circ$ denotes the cascade of neural networks, $\Phi_i$ is a multi-layer perceptron (MLP) with sine activation function in the $i$-th layer, and $W_P \in \mathbb{R}^{d \times 3}$ is the decoder to generate 3D point coordinates.

**Conditional Modulating.** While the point-wise generator can synthesize point clouds at arbitrary resolutions, it lacks class-conditional control, limiting its ability to generate category-specific data. To address this, we introduce a modulator $c : \mathbb{R}^d \to \mathbb{R}^d$, which is implemented as another MLP $\Psi$, to encode the label information and generate conditions for the point cloud generation:

$$c = \Psi_1 \circ \Psi_2 \circ \cdots \circ \Psi_L, \quad \Psi_i = \mathrm{ReLU}\left(m_i w_i' + b_i'\right), \quad (11)$$

where $\mathrm{ReLU}(\cdot) = \max(0, \cdot)$, $m_i \in \mathbb{R}^d$ denotes the conditional representations. The first layer input, $m_1$, is a one-hot matrix encoding class labels. Assume that there are $K$ classes in total, and each class has $N$ synthetic samples, then $m_1 \in \mathbb{R}^{KN}$ and $w_1' \in \mathbb{R}^{KN \times d}$.

The learned conditional representations are then used to modulate each layer of the generator, adjusting the frequency and phase features dynamically. The complete architecture is as follows:

$$g \odot c = [(\Psi_1 \odot \Phi_1) \circ (\Psi_2 \odot \Phi_2) \circ \cdots \circ (\Psi_L \odot \Phi_L)] W, \quad (12)$$

where $\odot$ is the element-wise multiplication. For clarity, in the following sections, we use $g(\epsilon, k)$ to denote $g \odot c$ with the $k$-th condition.

**Noise Distribution.** For noise sampling, we use uniform distribution instead of Gaussian distribution. The reasons are two-fold. First, INR requires inputs to be normalized within $[0, 1]$, which aligns naturally with the uniform distribution. Second, in the part segmentation task, each point must be assigned a label beforehand. A uniform distribution enables a straightforward division of noise samples accord-

ing to the category ratio, ensuring a balanced representation across different parts. See Figure 2 for an intuitive example.

### 4.3. Distillation Tasks

To comprehensively validate the effectiveness of DD3D, we conduct experiments on both the basic shape classification task and the challenging part segmentation task. Shape classification aims to assign each point cloud a label, emphasizing global information, while part segmentation predicts the label of each point, which is more fine-grained.

**Shape Classification.** The distillation objective of the shape classification task is defined as:

$$
\begin{aligned}
\mathcal{L}_{shape} = \sum_{k=1}^{K} \mathcal{D}(&\nabla\mathcal{L}_{cls}(f_\theta \circ r \circ g(\epsilon, k), Y_k^{\mathcal{S}}), \\
&\nabla\mathcal{L}_{cls}(f_\theta \circ r(B_k^{\mathcal{T}})), Y_k^{\mathcal{T}})),
\end{aligned}
\tag{13}
$$

where $K$ is the total classes of shapes, $B_k^{\mathcal{T}}$ and $Y_k^{\mathcal{T}}$ are a batch of real training data and labels.

**Part Segmentation.** In the part segmentation task, each shape is divided into multiple parts. For example, an airplane can be divided into its fuselage, wings, engines, and tail. Assigning these fine-grained labels before distillation helps stabilize the training process. Therefore, DD3D first partitions the noise into different segments based on its value. Then, the partitioned noise is fed into the generator and rotator to produce synthetic data. To effectively capture fine-grained details, DD3D aligns the gradients of each segment individually, rather than simply matching the gradients of the entire shape. This improves the preservation of local geometric features while maintaining overall structural coherence. A conceptual illustration is provided in Figure 2. The distillation of part segmentation task is formulated as:

$$
\begin{aligned}
\mathcal{L}_{part} = \sum_{k=1}^{K} \sum_{p \in k} \mathcal{D}(&\nabla\mathcal{L}_{seg}(f_\theta \circ r \circ (g\,(\epsilon, k) \odot M_p^{\mathcal{S}}), V_p^{\mathcal{S}}), \\
&\nabla\mathcal{L}_{seg}(f_\theta \circ r(B_k^{\mathcal{T}} \odot M_p^{\mathcal{T}}), V_p^{\mathcal{T}})),
\end{aligned}
\tag{14}
$$

where $p \in k$ indicates parts belonging to a shape, $V_p^{\mathcal{T}}, V_p^{\mathcal{S}}$ represents the real and synthetic part labels, and $M_p^{\mathcal{S}}, M_p^{\mathcal{T}}$ are the part-specific mask to extract gradients corresponding to each part. See Algorithm 1 for detailed descriptions.

### 4.4. Discussion

DD3D has demonstrated strong potential in capturing both global shape structures and local details in object-level point clouds. Visualizations in Section 4 further illustrate its effectiveness. However, applying DD3D to scene-level tasks, such as object detection, remains challenging. This limitation can be attributed to two key factors: First, scene-level tasks often involve a significant imbalance between

---

**Algorithm 1** DD3D for part segmentation

**Input:** Training dataset $\mathcal{T}$
**Ouput:** Model $f$, Rotator $r$, Generator $g$
**repeat**
  **for** $k = 1, \cdots, K$ **do**
    Sample a batch $B_k^{\mathcal{T}}, V_k^{\mathcal{T}} \sim \mathcal{T}$
    Sample noise $\epsilon \sim \mathcal{U}(0, 1)$
    Generate $V_k^{\mathcal{S}}, M_k^{\mathcal{S}}$ by partitioning noise $\epsilon$
    Generate point clouds $B_k^{\mathcal{S}} = g(\epsilon, k)$
    **for** $p \in k$ **do**
      Apply mask $M_p^{\mathcal{S}}, M_p^{\mathcal{T}}$ on $B_k^{\mathcal{S}}, B_k^{\mathcal{T}}$
      Compute $\nabla\mathcal{L}_{seg}^{\mathcal{S}}$ and $\nabla\mathcal{L}_{seg}^{\mathcal{T}}$
    **end for**
  **end for**
  Update $g$ with $\mathcal{L}_{part}$
  **repeat**
    Update $f, r$ with $\mathcal{L}_{seg}^{\mathcal{S}}$
  **until** inner-loop end
**until** outer-loop end

---

foreground and background points. Second, the detection task requires learning continuous bounding box coordinates, which cannot be predefined like segmentation labels, adding another layer of complexity.

## 5. Experiments

We benchmark our method on two fundamental tasks of point cloud analysis: shape classification (Section 5.1) and part segmentation (Section 5.2), followed by a series of analyses, including generalization (Section 5.3), ablation (Section 5.4), and visualization (Section 5.5).

**Datasets.** We employ three datasets of different scales for the shape classification task: $(i)$ ScanObjectNN (*OBJ_BG*) (Uy et al., 2019) is the smallest dataset but consists of real-world data, which is challenging to distillate. $(ii)$ ModelNet40 (Wu et al., 2015) is a larger synthetic dataset generated from CAD models. $(iii)$ MVPNet (Yu et al., 2023) is the largest dataset, containing 87K point clouds scanned from real-world videos. We use its subset MVPNet100, which includes data from the 100 most populous categories, to alleviate the influence of long-tail distribution, similar to the CAFIR-100 dataset. For the part segmentation task, we follow Qi et al. (2017a) and choose ShapeNet-part (Yi et al., 2016) dataset for evaluation. All the datasets use the standard data splits, and their detailed statistic information can be found in Appendix C.

**Data Preparation and Metrics.** Each cloud contains 1,024 points and is normalized into a unit sphere. We consider two settings: *Aligned* and *Rotated*. In the *Aligned* setting, both training and test point clouds have the same orienta-

Table 1: Shape classification results of different methods, mean accuracy (%) $\pm$ standard deviation. **Bold** indicates the best performance, and "-" means out-of-memory during distillation. CPC: Number of Clouds Per Class.

| Dataset | CPC | Coreset-based | | | Distillation-based | | | | Full Dataset |
|---|---|---|---|---|---|---|---|---|---|
| | | Random | Herding | K-Center | GM | DM | TM | DD3D | |
| ScanObjectNN (Aligned) | 1 | 22.00±2.56 | 16.29±1.37 | 18.18±1.04 | 26.34±2.07 | 25.90±1.34 | 26.42±2.08 | **30.62±1.75** | 66.96 |
| | 10 | 32.63±1.51 | 31.94±3.31 | 33.46±1.46 | 39.87±3.00 | 37.61±2.78 | 36.44±2.74 | **43.77±2.63** | |
| | 50 | 54.15±1.77 | 51.70±1.87 | 54.22±1.30 | 57.52±2.03 | 56.91±1.17 | - | **61.96±1.44** | |
| ScanObjectNN (Rotated) | 1 | 14.90±2.10 | 18.10±1.55 | 19.91±2.16 | 14.64±3.04 | 18.74±2.44 | 19.29±3.90 | **23.59±2.17** | 54.84 |
| | 10 | 20.50±1.26 | 20.20±2.19 | 22.05±1.76 | 20.55±3.99 | 20.26±4.31 | 19.20±4.52 | **25.84±3.11** | |
| | 50 | 42.98±1.84 | 43.39±1.34 | 44.29±2.07 | 47.74±1.82 | 48.11±2.30 | - | **50.26±1.42** | |
| ModelNet40 (Aligned) | 1 | 40.53±0.36 | 43.41±0.81 | 43.90±1.51 | 53.38±0.86 | 53.21±0.58 | 52.37±0.99 | **53.82±0.28** | 88.05 |
| | 10 | 71.89±0.29 | 74.63±0.48 | 73.13±0.78 | 75.45±0.82 | 74.45±0.47 | 75.39±1.32 | **76.31±0.49** | |
| | 50 | 82.37±0.45 | 82.75±0.49 | 82.73±0.28 | 81.74±0.55 | 83.02±1.16 | - | **83.91±0.23** | |
| ModelNet40 (Rotated) | 1 | 34.65±0.71 | 30.03±1.42 | 30.05±0.50 | 41.32±1.96 | 41.71±1.65 | 37.36±2.98 | **42.36±0.83** | 80.45 |
| | 10 | 58.87±0.65 | 56.03±0.62 | 57.69±0.97 | 55.69±1.63 | 55.45±1.80 | 56.21±1.14 | **58.14±1.36** | |
| | 50 | 70.13±0.64 | 70.02±0.71 | 69.68±0.59 | 68.92±0.73 | 69.31±0.79 | - | **71.27±0.32** | |
| MVPNet100 | 1 | 5.21±0.27 | 8.14±0.22 | 8.41±0.35 | 10.52±0.83 | 11.73±0.49 | 10.74±0.57 | **13.68±0.48** | 55.63 |
| | 10 | 15.99±0.30 | 22.11±0.21 | 20.54±0.21 | 25.68±0.77 | 25.71±0.69 | - | **31.14±1.31** | |
| | 50 | 30.14±0.27 | 35.87±0.24 | 35.48±0.44 | 37.41±0.57 | 36.83±0.20 | - | **40.61±0.38** | |

*Note*: All methods with rotated data are trained with the point cloud rotator. Ablations can be seen in Table 4.

Table 2: Part Segmentation results (%) on ShapeNet dataset.

| Ratio | Method | OA | Instance IoU | Class IoU |
|---|---|---|---|---|
| CPC=1 | Coreset | 61.24 | 48.21 | 31.61 |
| | GM | 65.56 | 50.98 | 33.96 |
| | DD3D | **73.06** | **60.27** | **37.73** |
| CPC=10 | Coreset | 77.78 | 65.03 | 48.19 |
| | GM | 78.32 | 65.79 | 49.88 |
| | DD3D | **80.37** | **66.70** | **50.59** |
| 100% | Full | 90.04 | 77.38 | 65.63 |

tion, while in the *Rotated* setting, both training and test data are rotated randomly. For the rotated data, we project them along the direction of maximum variance during preprocessing. Note that the point clouds in MVPNet only have 180° views, so we do not randomly rotate them. The details of pre-processing can be found in Appendix C. We report the Overall Accuracy (OA, %) of each method in the shape classification task and the average class intersection of union (IoU, %) in the part segmentation task.

**Baselines.** To demonstrate the effectiveness of our method, we choose two types of baselines: (1) Coreset-based methods, including Random, Herding (Welling, 2009) and K-Center (Sener & Savarese, 2018). (2) Distillation-based methods, including Gradient Matching (GM) (Zhao et al., 2021), Distribution Matching (DM) (Zhao & Bilen, 2023), and Trajectory Matching (TM) (Cazenavette et al., 2022). We choose GM as the distillation objection for DD3D as it

makes a trade-off between time and memory consumption. See Appendix D for the detailed hyperparameters.

**Backbones.** We provide a lightweight PointNet as the backbone, which abandons the transformation network because previous literature (Yu et al., 2024b) pointed out that complex network architecture may lead to degraded distillation performance. In the evaluation stage, we adopt various advanced backbones to evaluate the generalization ability of distilled datasets, including PointNet++ (Qi et al., 2017b), DGCNN (Wang et al., 2019), Point Transformer (Guo et al., 2021a), PointMLP (Ma et al., 2022), and PointNext (Qian et al., 2022). Results can be found in Table 3.

**Experimental Setup.** For each method, we perform the distillation process twice, evaluate each synthetic point cloud dataset five times (10 results in total), and report the mean and standard deviation. Baselines are all initialized with original data, while DD3D is trained from scratch. For the shape classification task, we consider three different distillation ratios with 1, 10, and 50 synthetic point clouds per class (CPC). For the part segmentation task, we choose CPC=1 and CPC=10 due to the limitation of GPU memory.

### 5.1. Shape Classification

The results of different methods on the shape classification task are shown in Table 1, from which we have the following observations. Firstly, the results of distillation-based methods consistently outperform coreset-based methods, demonstrating the effectiveness of DD. However, as the amount of synthetic data increases, the performance of the

Table 3: Cross-architecture results (%) with CPC=50.

| Dataset | Method | PointNet++ | DGCNN | PCT | PointMLP | PointNeXt |
|---|---|---|---|---|---|---|
| ScanObjectNN | DM | 56.02 | 51.47 | 52.72 | 51.33 | 51.82 |
| | GM | 55.38 | 52.98 | 53.28 | 51.33 | 52.81 |
| | DD3D | **57.14** | **53.36** | **54.04** | **52.50** | **53.36** |
| ModelNet40 | DM | 74.35 | 74.84 | 76.92 | 72.49 | 71.48 |
| | GM | 76.54 | 73.38 | 77.31 | 74.11 | 72.00 |
| | DD3D | **77.71** | **75.36** | **79.21** | **75.36** | **73.99** |
| MVPNet100 | DM | 33.20 | 31.26 | 33.92 | 32.58 | 31.17 |
| | GM | 31.35 | 29.88 | 31.43 | 31.79 | 30.82 |
| | DD3D | **34.19** | **32.94** | **35.82** | **33.08** | **32.75** |

Table 4: Ablation studies of the point cloud rotator.

| ModelNet40 (CPC=50) | Random | GM | DM | DD3D |
|---|---|---|---|---|
| PointNet | 14.75 | 9.47 | 10.16 | 17.91 |
| PointNet + PCA | 60.77 | 53.55 | 55.57 | 62.72 |
| PointNet + Rotator | 70.13 | 68.92 | 69.31 | 71.27 |
| Full Dataset | | 80.45 | | |

Table 5: DD3D under different resolutions.

| CPC=50 | Resolution | | | |
|---|---|---|---|---|
| | 256 | 512 | 1024 | Avg. |
| ScanObjectNN | 61.27 | 60.59 | **61.96** | 61.27 |
| ModelNet40 | 83.03 | 83.59 | **83.91** | 83.51 |
| MVPNet100 | 39.88 | 40.13 | **40.61** | 40.21 |

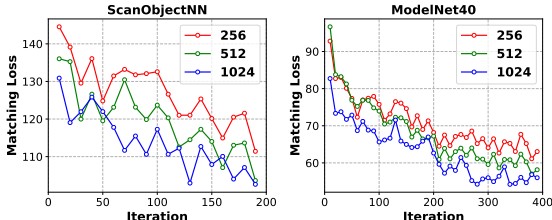

Figure 3: Matching losses of different resolutions.

coreset increases rapidly. Secondly, DD3D achieves state-of-the-art performance on all five datasets, demonstrating its superiority over traditional DD methods. Notably, DD3D obtains more improvements over baselines as the number of CPCs increases, possibly because the generator provides more diverse data. Thirdly, the results on the rotated data are weaker than those on the aligned data. Although we project the rotated data to the canonical orientation, *i.e.*, direction with maximum variance, these point clouds still have slightly different orientations, while the aligned data is manually registered, which is strictly towards the direction of gravity and therefore has better performance.

### 5.2. Part Segmentation

Table 2 presents the results of the part segmentation task on the ShapeNet dataset. Unlike classification, part segmentation requires learning both global shape structures and fine-grained part details, making it a more challenging task for DD. Since some traditional DD methods struggle with segmentation, we only compare DD3D against random coreset selection and GM. The results show that GM is slightly better than coreset selection as it is initialized by the real data. On the other hand, DD3D consistently outperforms both methods across all metrics by a large margin, demonstrating its effectiveness in learning the fine-grained features of point clouds. As expected, DD3D's performance does not yet reach the full dataset baseline. Nevertheless, it achieves 90% of the performance of the entire dataset using only 1% of the data, demonstrating its potential in 3D distillation.

### 5.3. Cross-architecture Generalization.

We evaluate whether DD3D can benefit different point cloud models. Specifically, we use PointNet as the distillation method and utilize five advanced point cloud models as evaluation methods, trained on the synthetic data from scratch.

Notably, we use synthetic data with CPC=50 to alleviate the randomness. The results are shown in Table 3, from which we can see that DD3D consistently outperforms DM and GM across different datasets and evaluation methods, proving that the synthetic data distilled by DD3D has better generalizability. This may be attributed to the generator that provides various point clouds in each epoch by sampling different noises, which plays a role like data augmentation. However, we can also observe that the results of evaluation methods are not as good as PointNet, emphasizing that the synthetic data is still biased by the distillation model.

### 5.4. Ablation Studies

**Point Cloud Rotator.** We first verify the effectiveness of the proposed point cloud rotator on the rotated ModelNet40 dataset. Specifically, we consider three different models: (1) PointNet, which is rotation-sensitive; (2) PointNet + PCA, which is rotation-invariant but sign-variant; (3) PointNet + Rotator, which is rotation- and sign-invariant. It can be observed from Table 4 that the performance of all methods drops rapidly when the data is randomly rotated. On the other hand, leveraging PCA to transform the point clouds into a canonical orientation can significantly improve the distillation performance. However, the results are still far from the model with the point cloud rotator, which reflects that sign ambiguity will seriously prevent the distillation model from learning meaningful synthetic data. Finally, it can be observed that the proposed rotator can help point cloud models to rotation-invariant representations, thus benefiting the learning of synthetic data.

**Point-wise Generator.** Next, we explore the performance of DD3D under different resolutions to verify the effectiveness and efficiency of the proposed generator. Typically, the shape classification task needs 1,024 points for training and

| Raw Images | DD3D | GM |
|---|---|---|

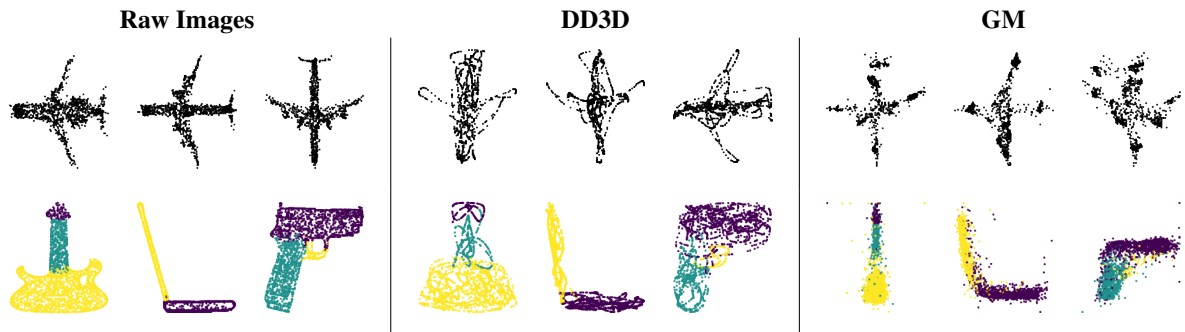

Figure 4: Visualizations of different methods. Top: ModelNet (Airplane). Bottom: ShapeNet (Guitar, Laptop, and Pistol).

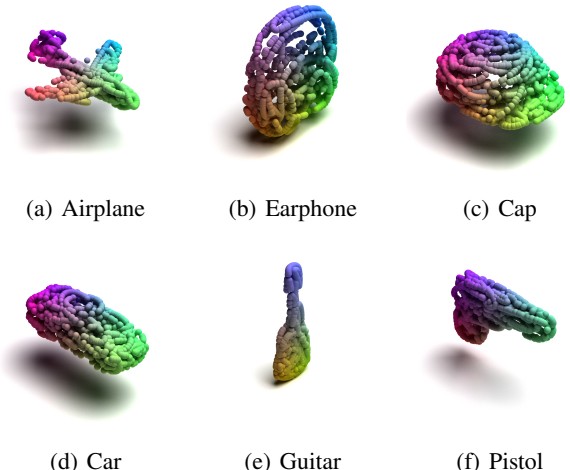

(a) Airplane    (b) Earphone    (c) Cap

(d) Car    (e) Guitar    (f) Pistol

Figure 5: Geometric details of points generated by DD3D.

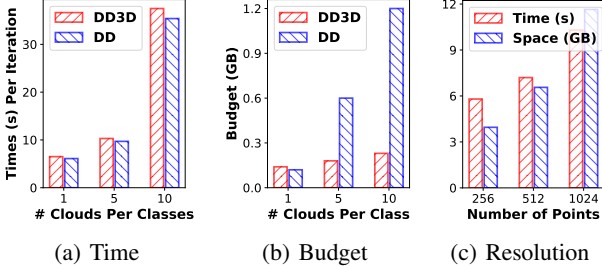

(a) Time    (b) Budget    (c) Resolution

Figure 6: Time and space overhead between DD and DD3D.

evaluation. In this experiment, we randomly sample 256 and 512 points from real data to supervise the distillation of DD3D. Once trained, we leverage DD3D to generate 1,024 points for evaluation. It is visible from Figure 3 that training on high-resolution point clouds can accelerate the convergence of DD3D but the final matching losses are similar. Moreover, Table 5 shows that different resolutions have similar performance. In some cases, low-resolution data also outperforms high-resolution point clouds, *e.g.*, ScanObjectNN. This discovery shows that DD3D can not only achieve stable results but also significantly reduce computational costs and GPU memory overhead.

## 5.5. Visualization

We visualize the real and synthetic point clouds in Figure 4 for a more intuitive comparison. The results of DD3D and GM are placed in the last two columns. It can be observed that the point clouds generated by GM tend to condense to some clusters, while some isolated points are left as noise. On the contrary, the point clouds generated by DD3D are coherent and encode the global geometric shapes. Moreover,

in ShapeNet, the point clouds of GM are squeezed, making its shape inconsistent with the real dataset, while the results of DD3D are more realistic and encode the spatial relationship between parts. Additionally, Figure 5 illustrates that DD3D not only preserves geometric details but also generates informative samples, further validating its effectiveness in 3D dataset distillation.

## 5.6. Time and Space Overhead

We compare the overhead between DD and DD3D from multiple views. Firstly, Figure 6(a) shows that the time overhead of DD3D is slightly higher than DD due to the generation of synthetic data. Then, we can observe from Figure 6(b) that the memory budget of DD grows faster than DD3D as the value of CPC increases. DD3D can save the budget of synthetic data by sharing the generator between different classes, and its memory is nearly 4x smaller than DD when CPC=10. Figure 6(c) illustrates the changes in time and space overhead of DD3D at different resolutions. We can see that training with low-resolution point clouds significantly reduces overhead, which is important for resource-constrained scenarios, such as edge computing.

## 6. Conclusion

This paper introduces DD3D for 3D point cloud distillation, which matches the rotation-invariant data distribution be-

tween real and synthetic data by transforming point clouds into a canonical orientation. Once trained, DD3D can synthesize point clouds at arbitrary resolutions, reducing memory budget and improving scalability. Extensive experiments on both classification and segmentation tasks validate the superiority of DD3D over traditional DD methods. A promising direction is to initialize DD3D with real data to improve its performance.

# Acknowledgment

This project is supported by the National Research Foundation, Singapore, under its Medium Sized Center for Advanced Robotics Technology Innovation.

# Impact Statement

This paper presents work whose goal is to advance the field of Machine Learning. In particular, it aims to accelerate the training of deep neural networks and reduce memory overhead. There are many potential societal consequences of our work, none of which we feel must be specifically highlighted here.

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

# A. Proof of Theorems

**Theorem A.1.** *Assume the classifier is a linear layer $W$ and $\mathcal{L}_{cls}$ can be simplified to the mean-squared error $\|XW - Y\|_F^2$. The objective of gradient matching is equal to variance preserving:*

$$\min_{\mathcal{S}} \mathcal{L}_{GM} = \min_{\mathcal{S}} \mathcal{D}\left(\nabla_W \mathcal{L}_{cls}^{\mathcal{S}}, \nabla_W \mathcal{L}_{cls}^{\mathcal{T}}\right) \quad \Rightarrow \quad \min_{\mathcal{S}} \left\|X_{\mathcal{S}}^\top X_{\mathcal{S}} - X_{\mathcal{T}}^\top X_{\mathcal{T}}\right\|_F^2, \tag{15}$$

*where $\mathcal{D}$ is a distance metric and $\nabla_W$ is the gradient with respect to $W$.*

*Proof.* The gradient of $\|XW - Y\|_F^2$ is denoted as $\nabla = X^\top(XW - Y)$. We can then match the gradients between the real and synthetic data:

$$\|\nabla_{\mathcal{S}} - \nabla_{\mathcal{T}}\|_F^2 = \|X_{\mathcal{S}}^\top(X_{\mathcal{S}}W - Y_{\mathcal{S}}) - X_{\mathcal{T}}^\top(X_{\mathcal{T}}W - Y_{\mathcal{T}})\|_F^2 \tag{16}$$

$$\leq \|W\|_F^2 \underbrace{\|X_{\mathcal{S}}^\top X_{\mathcal{S}} - X_{\mathcal{T}}^\top X_{\mathcal{T}}\|_F^2}_{\text{Variance}} + \underbrace{\|X_{\mathcal{S}}^\top Y_{\mathcal{S}} - X_{\mathcal{T}}^\top Y_{\mathcal{T}}\|_F^2}_{\text{Mean}}. \tag{17}$$

We can see that the first term is to preserve the variance of real data, and the second term aligns the average representations of samples belonging to the same class. These two terms can be combined if we set $\tilde{X}_{\mathcal{S}} = X_{\mathcal{S}} - X_{\mathcal{S}}^\top Y_{\mathcal{S}}$ and $\tilde{X}_{\mathcal{T}} = X_{\mathcal{T}} - X_{\mathcal{T}}^\top Y_{\mathcal{T}}$ for each class. Then we only need to match the variance between $\tilde{X}_{\mathcal{S}}$ and $\tilde{X}_{\mathcal{T}}$. $\square$

**Theorem A.2.** *Assume $X_{\mathcal{T}}$ follows a $d$-dimensional multivariate Gaussian distribution $\mathcal{N}(\boldsymbol{\mu}, \Sigma)$. Let $X_{\mathcal{T}}'$ be the rotated representations of $X_{\mathcal{T}}$ such that:*

$$\lambda_{max}\left(\mathbb{E}\left[{X_{\mathcal{T}}'}^\top X_{\mathcal{T}}'\right]\right) \leq \lambda_{max}\left(\mathbb{E}\left[X_{\mathcal{T}}^\top X_{\mathcal{T}}\right]\right) \quad \Rightarrow \quad \sigma_{max}(\mathbb{E}[X_{\mathcal{T}}']) \leq \sigma_{max}(\mathbb{E}[X_{\mathcal{T}}]), \tag{18}$$

*where $\lambda_{max}$ and $\sigma_{max}$ are the maximum eigenvalues and singular values, respectively.*

*Proof.* Firstly, the largest eigenvalue of the covariance matrix $X_{\mathcal{T}}^\top X_{\mathcal{T}}$ is equal to the largest singular value of $X_{\mathcal{T}}$. Therefore, we only prove the first inequality.

Secondly, as $X^\top X = \sum_{i=1}^n x_i^\top x_i$, for $X_{\mathcal{T}} \sim \mathcal{N}(\boldsymbol{\mu}, \Sigma)$, we have:

$$\mathbb{E}\left[X_{\mathcal{T}}^\top X_{\mathcal{T}}\right] = \mathbb{E}\left[\sum_{i=1}^n x_i^\top x_i\right] = \sum_{i=1}^n \mathbb{E}\left[x_i^\top x_i\right] = n\left(\boldsymbol{\mu}^\top \boldsymbol{\mu} + \Sigma\right), \tag{19}$$

$$\mathbb{E}\left[{X_{\mathcal{T}}'}^\top X_{\mathcal{T}}'\right] = \mathbb{E}\left[\sum_{i=1}^n R_i^\top x_i^\top x_i R_i\right] = \sum_{i=1}^n \mathbb{E}\left[R_i^\top x_i^\top x_i R_i\right] = \sum_{i=1}^n R_i^\top \mathbb{E}\left[x_i^\top x_i\right] R_i. \tag{20}$$

Thirdly, we have:

$$\lambda_{max}\left(\mathbb{E}\left[X_{\mathcal{T}}^\top X_{\mathcal{T}}\right]\right) = n\lambda_{max}\left(\boldsymbol{\mu}^\top \boldsymbol{\mu} + \Sigma\right), \tag{21}$$

$$\lambda_{max}\left(\mathbb{E}\left[{X_{\mathcal{T}}'}^\top X_{\mathcal{T}}'\right]\right) = \lambda_{max}\left(\sum_{i=1}^n R_i^\top \mathbb{E}\left[x_i^\top x_i\right] R_i\right) \leq \sum_{i=1}^n \lambda_{max}\left(R_i^\top \mathbb{E}\left[x_i^\top x_i\right] R_i\right) \tag{22}$$

$$= \sum_{i=1}^n \lambda_{max}\left(R_i^\top \boldsymbol{\mu}^\top \boldsymbol{\mu} R_i + R_i^\top \Sigma R_i\right) \leq \lambda_{max}\left(\mathbb{E}\left[X_{\mathcal{T}}^\top X_{\mathcal{T}}\right]\right). \tag{23}$$

The above inequality shows that the largest eigenvalue of $\mathbb{E}\left[X_{\mathcal{T}}^\top X_{\mathcal{T}}\right]$ is the upper bound of $\mathbb{E}\left[{X_{\mathcal{T}}'}^\top X_{\mathcal{T}}'\right]$. The equality holds if and only if the random rotation matrices are commutative, which is infeasible in practice. $\square$

# B. Implementation Details of DD3D

Here, we explain some details of DD3D, consisting of two important components: a point cloud rotator and a point-wise generator. Both components are built based on the SIREN (Sitzmann et al., 2020) model, which stacks multiple fully connected layers with $\sin(\cdot)$ activation to capture the high-frequency information. The PyTorch code is shown in Algorithm 2, where some details are highlighted.

**Algorithm 2** PyTorch code of DD3D

```
1   import torch
2   import torch.nn as nn
3   import SIREN
4
5   class Rotator(nn.Module):
6       def __init__(self, hidden_dim, w0):
7           super().__init__()
8
9           # w0 is to adjust the frequency of sine function
10          self.sign_encoder = SIREN(1, hidden_dim, w0=w0)
11          self.sign_decoder = SIREN(hidden_dim, 1, w0=1.)
12
13      def forward(self, x):
14          x = x.unsqueeze(-1) # x: [B, N, 3, 1]
15
16          feat = self.sign_encoder(x).mean(dim=1, keepdim=True) # [B, N, 3, 1] -> [B, 1, 3, d]
17          feat = self.sign_decoder(feat) # [B, 1, 3, d] -> [B, 1, 3, 1]
18          sign = torch.sign(feat) # sign-equivariant
19
20          x = x * sign # [B, N, 3, 1] * [B, 1, 3, 1] -> [B, N, 3, 1]
21          return x.squeeze(-1)
22
23
24  class ConditionalGenerator(nn.Module):
25      def __init__(self, genetator, num_classes, cpc, condition_dim, num_layers):
26          super().__init__()
27
28          self.genetator = genetator
29          self.lookup = nn.Embedding(num_classes * cpc, condition_dim) # class index as condition
30          self.num_layers = num_layers
31
32          self.layers = nn.ModuleList([])
33
34          for _ in range(self.num_layers - 1):
35              self.layers.append(nn.Sequential(nn.Linear(condition_dim, condition_dim), nn.ReLU()))
36
37      def forward(self, noise, class_indices):
38
39          # noise [B, N, 1]
40          # class_inices [B, C]
41
42          mod = self.lookup(class_indices)
43          mods = [mod]
44
45          for layer in self.layers:
46              mod = layer(mod)
47              mods.append(mod)
48
49          return self.genetator(noise, tuple(mods))
```

## C. Details of Datasets

The detailed statistical information of the datasets used in this paper is shown in Table 6. We list the sources of the datasets and their licenses in the following.

- **ScanObjectNN**: https://github.com/feiran-l/rotation-invariant-pointcloud-analysis

- **ModelNet40**: http://modelnet.cs.princeton.edu/ModelNet40.zip

- **MVPNet**: https://github.com/GAP-LAB-CUHK-SZ/MVImgNet

- **ShapeNet**: https://github.com/feiran-l/rotation-invariant-pointcloud-analysis

## D. Hyperparameters

The hyperparameters of baselines and DD3D are listed in Tables 7 and 8, respectively.

Table 6: Details of datasets

|  | ScanObjectNN | ModelNet40 | MVPNet | ShapeNet |
|---|---|---|---|---|
| # Shape Classes | 15 | 40 | 100 | 16 |
| # Part Classes | - | - | - | 50 |
| # Training Samples | 2,322 | 9,843 | 62,494 | 14,007 |
| # Validation Samples | 580 | 2,468 | 15,670 | 2,874 |
| Resolution | 1,024 | 1,024 | 1,024 | 2,048 |

Table 7: Hyperparameters used for Data Synthesis.

|  | ScanObjectNN | ModelNet40 | MVPNet100 | ShapeNet |
|---|---|---|---|---|
| Optimizer | Adam | Adam | Adam | Adam |
| Initial LR | 0.001 | 0.001 | 0.001 | 0.001 |
| Batch Size | 32 | 32 | 64 | 32 |
| Iterations | 200 | 400 | 600 | 200 |
| Weight Decay | 0.0005 | 0.0005 | 0.0005 | 0.0005 |
| Augmentation | Scale, Jitter, Rotate | Scale, Jitter, Rotate | Scale, Jitter, Rotate | Scale, Jitter, Rotate |
| Scheduler | StepLR (Decay 0.1 / 100 iter) | StepLR (Decay 0.1 / 100 iter) | StepLR (Decay 0.5 / 200 iter) | - |

Table 8: Hyperparameters used for Validation.

|  | ScanObjectNN | ModelNet40 | MVPNet100 | ShapeNet |
|---|---|---|---|---|
| Optimizer | Adam | Adam | Adam | Adam |
| Initial LR | 0.001 | 0.001 | 0.001 | 0.001 |
| Batch Size | 8 | 8 | 32 | 8 |
| Epochs | 200 | 200 | 200 | 200 |
| Weight Decay | 0.0005 | 0.0005 | 0.0005 | 0.0005 |
| Augmentation | Scale, Jitter, Rotate | Scale, Jitter, Rotate | Scale, Jitter, Rotate | Scale, Jitter, Rotate |
| Scheduler | StepLR (Decay 0.1 / 100 epoch) | StepLR (Decay 0.1 / 100 epoch) | CosineAnnealingLR | - |

