# OpenReview forum: "Point Cloud Dataset Distillation"
_ICML.cc/2025/Conference — ICML 2025 poster_

### Official Review · Reviewer_z87V · 2025-03-10

**Overall Recommendation:** 3

**Summary:**

This paper studies the dataset distillation for point cloud. This paper claims this is the first study on dataset distillation for point cloud, aiming at the two challenges: diverse orientations and resolutions in 3D space. To overcome these issues, this paper 1) proposes a plug-and-play point cloud rotator to align the point cloud to a canonical orientation; 2) devises a point-wise generator
to produce point clouds at various resolutions based on the sampled noise amount.  The experimental results demonstrate the proposed method achieves higher score than previous methods.


### Update after rebuttal

The authors addressed my concerns. After reading other reviews, I keep my initial rating "weak accept".

**Claims And Evidence:**

yes

**Essential References Not Discussed:**

No

**Experimental Designs Or Analyses:**

no issues.

**Methods And Evaluation Criteria:**

yes

**Other Comments Or Suggestions:**

-

**Other Strengths And Weaknesses:**

Strengths:

This paper makes the first attempt for point cloud dataset distillation, which might provide some insights for the community.

The proposed method seems to be efficient in comparison with previous methods.

Weaknesses:

=== I am not familiar with this task. ===

- During the dataset distillation process, the task model is also trained on the original dataset. Therefore, how does DD significantly reduce the computational cost of training neural networks from scratch (LINE09, PAGE01)?

- After the distillation process, do we need to further train another task model on the distilled dataset?

- For part segmentation (L338), "each
shape is divided into multiple parts. For example, an airplane
can be divided into its fuselage, wings, engines, and
tail." For other objects, how to divide the parts? The division strategy impacts the performance and how?

- The performance is low, which impacts the practical application. For example, for modelnet40, current SOTA has achieved over 94% accuracy, while the highest score is 83% in this paper. For part segmentation, the performance lags far behind.

**Questions For Authors:**

-

**Relation To Broader Scientific Literature:**

Dataset distillation has not yet been studied in the community. This paper, as the first attempts, might be helpful for the community.

**Theoretical Claims:**

do not check carefully.

---

> ### Author Rebuttal · Authors · 2025-03-31
>
> We are grateful for your constructive advice and the opportunity to address your concerns.
>
> &nbsp;
> ### ``Other Strengths And Weaknesses``
>
> > **Q1: The task model is also trained on the original dataset. How does DD significantly reduce the computational cost?**
>
> A1: We would like to clarify that the task model is not trained on the original datasets. Generaaly, gradient matching employs a bi-level optimization framework:
> - Inner-loop: Optimize task model using **synthetic data**.
> - Outer-loop: Fix the task model and match the gradients between a batch of real and synthetic data.
>
> After distillation, the synthetic data contains the gradient information of real data, thereby converging the models with fewer iterations, i.e., batches.
>
> > **Q2: Do we need to further train another task model on the distilled dataset?**
>
> A2: Yes, another task model needs to be trained from scratch. Although the task model is optimized in the inner loop, its performance is far from ideal, as the synthetic data is not fully trained.
>
> One benefit is that well-trained synthetic data can be used to train different task models. For example, we can use PointNet during distillation and train more complex models, such as DGCNN or Point Transformer, after distillation. A cross-architecture performance is reported in ``Table 3`` of the main text.
>
> > **Q3: For other objects, how to divide the parts? The division strategy impacts the performance and how?**
>
> A3: The parts of objects are defined by the labels of the real datasets. For example, the ShapeNetCore dataset contains 16 object classes and 50 part classes in total. There is a segmentation dictionary to store the correspondence between objects and parts, like {'Airplane': [0, 1, 2, 3], 'Bag': [4, 5], ...}.
>
> In summary, **the division strategy is fixed and depends on real data**, which will not have different impacts on the performance of DD3D.
>
> > **Q4: Performance of DD3D**
>
> A4: We understand your concerns about the performance of DD. We propose two strategies to mitigate the performance gap between models trained on real and synthetic data
>
> 1. Improving the number of CPC. We add the performance of DD3D with CPC=100 in ModelNet40 and ScanObject, and find that it can achieve comparable performance with the full dataset.
> | ModelNet40 | CPC=50 | CPC=100 | | ScanObject | CPC=50 | CPC=100 |
> | :--: | :--: | :--: | :--: | :--: | :--: | :--: |
> | GM | 81.74 | 84.17 | | GM | 57.52 | 62.82 |
> | DD3D | 83.91 | **86.68** | | DD3D | 61.96 | **65.51** |
> | Full | 88.05 | 88.05 | | Full | 66.96 | 66.96 |
>
> 2. Combined with knowledge distillation (KD). Recent DD methods suggest that leveraging KD can significantly improve the performance of DD. We can also adopt this strategy to achieve nearly lossless performance.
> | ModelNet40 | CPC=50 | CPC=100 | | ScanObject | CPC=50 | CPC=100 |
> | :--: | :--: | :--: | :--: | :--: | :--: | :--: |
> |DD3D | 83.91 | 86.68 | |DD3D | 61.96 | 65.51 |
> |DD3D+KD | 86.55 | **88.75** | |DD3D+KD | 65.06 | **66.84** |
> | Full | 88.05 | 88.05 | | Full | 66.96 | 66.96 |
>
> > **Q5: Comparison with state-of-the-art point cloud analysis models.**
>
> A5: The performance of DD3D is behind the SOTA point cloud analysis models, as we only adopt the vanilla 3-layer PointNet as the distillation network for DD3D, which makes a good trade-off between effectiveness and efficiency.
>
> Notably, in the image distillation task, DD methods often use a 3-layer ConvNet as the distillation network [1]. We report the performance of vision DD (Tiny-ImageNet) and point cloud DD (ModelNet40) below.
>
> | Tiny-ImageNet | 3-layer ConvNet | ResNet-18 |
> | :--: | :--: | :--: |
> | IPC=50 | 37.6% | 59.8% |
>
> | ModelNet40 | 3-layer PointNet | PointNext |
> | :--: | :--: | :--: |
> | CPC=50 | 88.1% | 94.00% |
>
> We can observe that the performance of the 3-layer ConvNet is far from the ResNet-18 model. But it still promotes the development of vision DD. On the other hand, the performance of the 3-layer PointNet is close to the SOTA models, like PointNext.
> We believe that our research will also benefit the 3D DD field, and future works will extend DD to the SOTA point cloud analysis models.
>
> [1] Dataset Condensation with Gradient Matching. ICLR 2021.

---

### Official Review · Reviewer_yUbr · 2025-03-13

**Overall Recommendation:** 3

**Summary:**

In this paper, the authors presented DD3D, a novel framework for 3D point cloud distillation that aligns the rotation-invariant data distribution between real and synthetic data by transforming point clouds into a canonical orientation. Once trained, DD3D is capable of synthesizing point clouds at arbitrary resolutions, thereby significantly reducing memory consumption and enhancing scalability. Through extensive experiments on both shape classification and part segmentation tasks, the proposed method can achieve the superior performance compared to conventional dataset distillation methods.

**Claims And Evidence:**

The proposed method opens up new opportunities for efficient training of point cloud models and broadens the applicability of dataset distillation techniques to unstructured 3D data, compared to existing distillation methods have primarily focused on structured data such as images, videos, and text.

**Essential References Not Discussed:**

Not applicable.

**Experimental Designs Or Analyses:**

The authors conduct comprehensive evaluations of DD3D across diverse datasets and scenarios, demonstrating that it consistently outperforms traditional distillation methods.

To reinforce the author’s argument, it would be beneficial to compare the performance and feature characteristics of models trained on both aligned and unaligned orientations. Such a comparison would provide clearer insights into the effect of rotation on feature learning and model robustness.

**Methods And Evaluation Criteria:**

One of the core idea is that rotation invariance positively contributes to the effectiveness of distillation. In this context, I am curious about the authors’ perspective on the potential impact of incorporating more recent rotation-invariant or rotation-equivariant models, and how such integration might influence the performance or generalizability of the proposed approach.

While some performance drop compared to using the full dataset is expected in dataset distillation, it remains a notable limitation for high-accuracy applications. Despite the improvements over existing methods, concerns remain about potential degradation when scaling to larger or more complex tasks.

**Other Comments Or Suggestions:**

The authors should revise several typos and grammatical errors in the paper. (not critical)

**Other Strengths And Weaknesses:**

I mentioned all comments including reasons and suggestions in each section and I recommend that the author will provide my concerns in the rebuttal.

**Questions For Authors:**

Rather than posing additional questions, I would like the authors to carefully review and address the concerns I have raised above in detail.

**Relation To Broader Scientific Literature:**

The authors present a theoretical analysis showing that matching rotation-invariant features is essential for effective 3D point cloud distillation. They demonstrate that random rotations weaken the principal components of real data, leading to degraded distillation performance, which strongly justifies the use of the proposed rotator.

**Theoretical Claims:**

The paper provides solid mathematical justifications that effectively support the proposed approach and offer clear insights into why the techniques work as intended.

---

> ### Author Rebuttal · Authors · 2025-04-01
>
> Thanks for the detailed and helpful comments. We reply to the comments in detail. Hope to address your concerns.
>
> &nbsp;
> ### ``Methods And Evaluation Criteria``
>
> > **Q1: Potential impact of incorporating more recent rotation-invariant or rotation-equivariant models.**
>
> A1: Thanks for the great advice. We have tried several rotation-invariant or equivariant models for DD. However, there are two issues.
>
> 1. The architectures of advanced rotation-invariant models are quite different, making it hard to transfer to rotation-variant models.
> 2. These models often involve complex components. A recent survey on DD [1] shows that more complex models do not necessarily lead to better performance. See Q2 for a performance comparison.
>
> On the other hand, the rotator of DD3D can alleviate the above two issues as it is lightweight and **model-agnostic**.
>
> [1] Dataset distillation: A comprehensive review. TPAMI 2024.
>
> > **Q2: How such integration might influence the performance or generalizability of the proposed approach?**
>
> A2: We compare DD3D with some SOTA rotation-invariant methods, including RISurConv, TetraSphere (suggested by Reviewer vcTR), and SGMNet (suggested by Reviewer gVCR). The results are shown below.
> | ModelNet40 | CPC=10 | CPC=50 | Full |
> | :-- | :--: | :--: | :--: |
> | DD3D | 58.14 | **71.27** | 80.45 |
> | RISurConv | 55.81 | OOM | **95.60** |
> | TetraSphere | 54.72 | OOM | 90.50 |
> | SGMNet | **58.33** | 70.48 | 80.77 |
>
> We have the following observations:
> 1. Both RISurConv and TetraSphere perform well in the full dataset training. However, they are not suitable for the DD task, as they all need to build a k-NN graph to learn rotation-invariant representations. During distillation, the synthetic point clouds are dynamically optimized, resulting in different nearest neighbors. As a result, we need to **re-calculate the k-NN graph** in each iteration, which significantly increases the time and space overhead.
> 2. On the other hand, the rotator of DD3D and SGMNet are model-agnostic, which can be easily combined with different point cloud models. Here we use the 3-layer PointNet as the backbone, and their results are better than RISurConv and TetraSphere. This indicates that the complex models do not always lead to better distillation performance. In contrast, simple models may be more suitable for DD.
>
> > **Q3: Performance drop compared to using the full dataset and limitation for high-accuracy applications.**
>
> A3: We fully understand your concerns about the accuracy. For applications requiring high accuracy, we recommend two ways to improve the accuracy of DD3D.
>
> 1. Improving the number of CPC. We report the performance of DD3D with CPC=100 in ModelNet40 and ScanObject, and find that it can achieve comparable performance with the full dataset.
>
> | ModelNet40 | CPC=50 | CPC=100 | | ScanObject | CPC=50 | CPC=100 |
> | :--: | :--: | :--: | :--: | :--: | :--: | :--: |
> | GM | 81.74 | 84.17 | | GM | 57.52 | 62.82 |
> | DD3D | 83.91 | **86.68** | | DD3D | 61.96 | **65.51** |
> | Full | 88.05 | 88.05 | | Full | 66.96 | 66.96 |
>
> 2. Combined with knowledge distillation (KD). Recent DD methods suggest that leveraging KD can significantly improve the performance of DD. We can also adopt this strategy to achieve nearly lossless performance.
>
> | ModelNet40 | CPC=50 | CPC=100 | | ScanObject | CPC=50 | CPC=100 |
> | :--: | :--: | :--: | :--: | :--: | :--: | :--: |
> |DD3D | 83.91 | 86.68 | |DD3D | 61.96 | 65.51 |
> |DD3D+KD | 86.55 | **88.75** | |DD3D+KD | 65.06 | **66.84** |
> | Full | 88.05 | 88.05 | | Full | 66.96 | 66.96 |
>
> > **Q4: Potential degradation when scaling to larger or more complex tasks.**
>
> A4: We conduct semantic segmentation experiments on the scene-level S3DIS dataset, which contains more than 80M point clouds for training. Generally, the scene-level task is more challenging than the object-level task, as it includes more data and noise. We can observe that DD3D consistently outperforms GM by a large margin, demonstrating its effectiveness in large-scale scene-level tasks.
> | S3DIS (OA / mIoU) | GM | DD3D | Full |
> | :-- | :--: | :--: | :--: |
> | CPC=1 (0.06%) | 53.09% / 0.3674 | **57.45% / 0.4043** | 73.78% / 0.5786 |
> | CPC=10 (0.6%) | 61.72% / 0.4438 | **64.05% / 0.4624** | 73.78% / 0.5786 |
>
> &nbsp;
> ### ``Experimental Designs Or Analyses``
>
> > **Q5: Compare the performance and feature characteristics of models trained on both aligned and unaligned orientations.**
>
> A5: ``Table 4`` in the main text shows the performance of models trained on both aligned and unaligned orientations. We can see that the performance of GM and DM drops quickly without the help of the rotator of DD3D, demonstrating the effectiveness of DD3D in distillating 3D datasets.
>
> &nbsp;
> ### ``Other Comments Or Suggestions``
>
> > **Q6: The authors should revise several typos and grammatical errors in the paper.**
>
> A6: We will carefully polish our paper and revise typos.

---

### Official Review · Reviewer_bmGW · 2025-03-15

**Overall Recommendation:** 3

**Summary:**

The paper introduces DD3D, a dataset distillation method tailored for 3D point clouds, addressing challenges of orientation diversity and varying resolutions. It proposes a rotation-invariant feature matching approach, a point cloud rotator for canonical alignment, and a point-wise generator that efficiently produces multi-resolution point clouds. Extensive experiments demonstrate DD3D’s effectiveness in shape classification and part segmentation, showing strong cross-architecture and cross-resolution generalization while reducing memory usage.

**Claims And Evidence:**

Yes

**Essential References Not Discussed:**

No

**Experimental Designs Or Analyses:**

Yes

**Methods And Evaluation Criteria:**

Yes

**Other Comments Or Suggestions:**

It seems that the link in the paper is not anonymous.

**Other Strengths And Weaknesses:**

Strengths:

1, The paper is well-written and easy to understand.

2, It is the first work to introduce dataset distillation for 3D point clouds.

3, The rotation-equivariant design for point cloud distillation is intuitive and well-motivated.

Weaknesses:

1, While I am not an expert in this area, I am familiar with other data-efficient methods such as knowledge distillation and semi-supervised learning, and dataset distillation appears to demonstrate promising results in comparison.

2, However, the performance improvements over other methods not specifically designed for 3D tasks seem relatively modest, raising concerns about the method’s overall impact.

**Questions For Authors:**

No

**Relation To Broader Scientific Literature:**

No

**Theoretical Claims:**

Yes

---

> ### Author Rebuttal · Authors · 2025-03-31
>
> Thank you for your thoughtful review and suggestions.
>
> &nbsp;
> ### ``Other Strengths And Weaknesses``
>
> > **Q1: Comparison between dataset distillation,  knowledge distillation, and semi-supervised learning.**
>
> A1: Dataset distillation (DD) and knowledge distillation (KD) are two orthogonal directions in efficient deep learning. DD aims to compress data, and KD is used to compress models. A recent DD method [1] suggests that incorporating KD into DD can alleviate the performance gap between the real and synthetic data. To verify this, we report the performance of **DD3D** and **DD3D+KD** in the ModelNet40 and ScanObject datasets below.
>
> | ModelNet40 | CPC=50 | CPC=100 | | ScanObject | CPC=50 | CPC=100 |
> | :--: | :--: | :--: | :--: | :--: | :--: | :--: |
> |DD3D | 83.91 | 86.68 | |DD3D | 61.96 | 65.51 |
> |DD3D+KD | 86.55 | **88.75** | |DD3D+KD | 65.06 | **66.84** |
> | Full | 88.05 | 88.05 | | Full | 66.96 | 66.96 |
>
> We can see that with the help of KD, the performance of synthetic datasets is close to the real datasets, indicating the potential of combining DD and KD. Moreover, semi-supervised learning is also a potential application of DD. Leveraging the knowledge of a few labeled samples to compress the massive unlabeled data is also a promising direction.
>
> [1] Squeeze, Recover and Relabel: Dataset Condensation at ImageNet Scale From A New Perspective. NeurIPS 2023.
>
> > **Q2: Performance improvements over other methods not specifically designed for 3D tasks seem relatively modest.**
>
> A2: The advantages of DD3D over other methods not specifically designed for 3D tasks, e.g., GM, DM, and TM, concentrate on addressing the rotation and resolution issues.
>
> 1. For the randomly rotated datasets, DD3D outperforms other methods by a large margin. We report the results in Table 4 of the main text below. We can see that the performance of GM and DM drops quickly without the help of the rotator of DD3D, demonstrating the effectiveness of DD3D in distillating 3D datasets.
> | ModelNet40 | Random | GM | DM | DD3D |
> | :-- | :--: | :--: | :--: | :--: |
> | PointNet | 14.75 | 9.47 | 10.16 | 17.91 |
> | PointNet + PCA | 60.77 | 53.55 | 55.57 | 62.72 |
> | PointNet + Rotator | 70.13 | 68.92 | 69.31 | **71.27** |
>
> 2. For the large-scale datasets, DD3D enables “low-resolution training and high-resolution generation”, which significantly reduces the space overhead during distillation. We report the space overhead of GM and DD3D below, from which we can observe that the memory cost of GM is 10x higher than DD3D, but its performance is still not as good as DD3D. Therefore, DD3D can handle large-scale datasets and larger CPC.
> | CPC (1/5/10) | Memory (MB) | Performance |
> | :-- | :--: | :--: |
> | GM | 120 / 1200 / 6000 | 53.38 / 72.11 / 75.45 |
> | DD3D | 140 / 230 / **630** | 53.82 / 73.54 / **76.31** |
>
> &nbsp;
> ### ``Other Comments Or Suggestions``
>
> > **Q3: It seems that the link in the paper is not anonymous.**
>
> A3: We kindly clarify that this link is not related to our paper (our code is in the Supplementary Material) but to a related work on point cloud DD. **This link does not violate the double-blind reviewing policy.**

---

### Official Review · Reviewer_gVCR · 2025-03-18

**Overall Recommendation:** 3

**Summary:**

The paper addresses the problem of distilling large 3D point cloud datasets into a much smaller set while preserving model performance. The paper identifies two key challenges unique to point clouds: random orientation and variable resolution. To tackle these issues, the authors propose a novel framework called DD3D, which introduces a plug-and-play point cloud rotator that aligns each point cloud to a canonical orientation, making the distilled data rotation-invariant. It also uses a point-wise generator to produce point clouds from noise, allowing flexible output sizes (arbitrary resolutions) while training on lower resolutions. The rotator and generator are optimized jointly using a gradient-matching distillation objective so that models trained on the small synthetic set achieves performance close to ones trained on the full set. The paper demonstrates DD3D on 3D shape clasification and part segmentation, and shows competitive performance through this dataset distillation process.

**Claims And Evidence:**

The claims made in the submission are supported by clear and convincing evidences.

**Essential References Not Discussed:**

The references are adequate.

**Experimental Designs Or Analyses:**

The experimental designs are valid.

**Methods And Evaluation Criteria:**

The proposed methods make sense for the problem.

**Other Comments Or Suggestions:**

See Other Strengths And Weaknesses.

**Other Strengths And Weaknesses:**

Strengths
- This work is the first to focus on dataset distillation for 3D point clouds​, an extension of data distillation methods beyond images. It clearly pinpoints unique challenges in 3D that were not handled by prior 2D distillation methods​. The problem is well-motivated by practical needs (reducing memory/training costs on huge 3D datasets).
- The paper provides a theoretical analysis to justify the approach. While intuitive, the paper proves (under simplifying assumptions) that the gradient-matching distillation objective is equivalent to preserving the data variance, and that random rotations can weaken the principal components (variance) of the data, harming distillation​. This analysis highlights why matching rotation-invariant features is important.
- The paper backs up its claims with comprehensive experiments on multiple datasets and tasks, including 3D point cloud classification and segmentation. Notably, the authors tested performance on cross-architecture generalization, and show general applicability of the distillation scheme.

Minor Weaknesses
- Since some point cloud models are inherently rotation-invariant (e.g. [1, 2], and should be many more recent methods), comparing DD3D to a scenario where a rotation-invariant model used (instead of adding an external rotator) could help delineate the benefits of the proposed approach. Any discussion in this direction is welcomed.
- The framework adds extra components (the rotator and generator) to the distillation pipeline, which increases the method’s complexity. The paper would benefit from more details on these components. For instance, the authors could elaborate on the point-wise generator’s architecture and training process, e.g., do they simply adjust the amount of input noise, how is the generator supervised across different point counts, etc. Likewise, the rotator module is a key piece of the pipeline- a deeper understanding of its training procedure or potential failure cases (e.g., ambiguous alignments for symmetric shapes) would be useful. Without these details, it’s hard to assess the overall robustness under different input settings.

[1] Li, Xianzhi, et al. "A rotation-invariant framework for deep point cloud analysis." IEEE transactions on visualization and computer graphics 28.12 (2021): 4503-4514.
[2] Xu, Jianyun, et al. "Sgmnet: Learning rotation-invariant point cloud representations via sorted gram matrix." Proceedings of the IEEE/CVF International Conference on Computer Vision. 2021.

**Questions For Authors:**

See Other Strengths And Weaknesses.

**Relation To Broader Scientific Literature:**

The paper could be significant for efficient 3D network training and provide insights in this field.

**Theoretical Claims:**

I have checked the theoretical claims in the main paper and saw no obvious errors.

---

> ### Author Rebuttal · Authors · 2025-04-01
>
> We sincerely appreciate your thoughtful feedback and insightful questions.
>
> &nbsp;
> ### ``Other Strengths And Weaknesses``
>
> > **Q1: Comparing DD3D to a scenario where a rotation-invariant model is used could help delineate the benefits of the proposed approach. Any discussion in this direction is welcomed.**
>
> A1: As suggested, we compare the performance between DD3D, SGMNet [1], and Li et al [2] in the randomly rotated ModelNet40 dataset. Specifically, DD3D and SGMNet are model-agnostic methods, and we use our 3-layer PointNet as the backbone. The results are shown below.
> | ModelNet40 | CPC=10 | CPC=50 | Full |
> | :-- | :--: | :--: | :--: |
> | DD3D | 58.14 | **71.27** | 80.45 |
> | SGMNet | **58.33** | 70.48 | 80.77 |
> | Li et al | 43.82 | OOM | **89.4** |
>
> We have the following observations:
> 1. DD3D has a similar performance to SGMNet, demonstrating the effectiveness of the proposed rotator. Generally, DD3D leverages the fact that eigenvectors are rotation-equivalent to alleviate the influence of random rotations, i.e., $(PR)(R^\top U)=PU$. SGMNet uses the outer product to eliminate rotations, i.e., $(PR)(PR)^\top=PP^\top$. Compared to SGMNet, DD3D only needs to maintain a $3 \times 3$ matrix $U$ rather than an $N \times N$ matrix $XX^\top$, which is more efficient.
> 2. While Li et al’s work has the highest accuracy in the full dataset, it does not perform well in the distillation task. This observation is consistent with a recent work in DD [3] that networks with complex components may affect the performance of DD.
> 3. Li et al’s work is out-of-memory in the CPC=50 setting, due to the quadratic complexity of fastest sampling. Each time the synthetic point cloud is optimized, the fastest sampling needs to be recalculated, which significantly increases the time and memory overhead of the distillation process.
>
> Based on the above observation, we have the following discussion:
> 1. **Distillation network is important**. The SOTA point cloud models often need to build a k-NN graph or apply fastest sampling on the point clouds, which is not suitable for DD as it costs too much time and space.
> 2. **Model-agnostic v.s. Model-specific**. Li et al’s work is model-specific and achieves the highest accuracy. However, it is infeasible to transfer it to some simple models, like PointNet. On the other hand, SGMNet is model-agnostic and can be used for different models, which is more suitable for DD.
>
> > **Q2: The framework adds extra components (the rotator and generator) to the distillation pipeline, which increases the method’s complexity.**
>
> A2: Compared to the gradient matching process, the rotator and generator only slightly increase the computational complexity. To verify this conclusion, we use `line_profiler` to record the time used in rotator, generator, and gradient matching, respectively, and report their percentages. The results below indicate that the rotator and generator only slightly impact the computational overhead:
>
> | CPC | Rotator (%) | Generation (%) | Matching (%) |
> | :---: | :---: | :---: | :---: |
> | 1     | 0.5 | 3.9 | 95.6 |
> | 10   | 0.3 | 2.7 | 97.0 |
> | 50   | 0.1 | 1.0 | 98.9 |
>
> > **Q3: The authors could elaborate on the point-wise generator’s architecture and training process.**
>
> A3: Thanks for the great advice. We will add more description of the rotator and generator of DD3D in the revision. We briefly introduce the training process of the generator below.
>
> - The point-wise generator $\mathbb{R} \rightarrow \mathbb{R}^3$ aims to map each sampled noise into a point. Therefore, we can adjust the amount of input noise to control the resolution of synthetic point clouds.
> - During training, we sample fewer points from the real data and calculate their gradient as supervision to update the generator. During inference, we can sample more noise to ensure geometric details.
>
> > **Q4: A deeper understanding of its training procedure or potential failure cases (e.g., ambiguous alignments for symmetric shapes) would be useful.**
>
> A5: We will add more visualization in the revision to illustrate how the rotations affect the training process of synthetic datasets.

---

### Official Review · Reviewer_vcTR · 2025-03-25

**Overall Recommendation:** 3

**Summary:**

This paper proposes DD3D, the first dataset distillation method designed specifically for 3D point cloud data. DD3D addresses two critical challenges in point cloud distillation: orientation misalignment and varying resolutions. The authors first establish theoretically that an ideal dataset distillation should preserve the intrinsic variance of the dataset and that orientation-misaligned samples introduce undesirable perturbations. To solve this, they introduce a novel plug-and-play rotator to consistently align point clouds to canonical orientations, resolving rotation and sign ambiguities. Additionally, DD3D employs a point-wise generator capable of synthesizing high-resolution point clouds from low-resolution training samples, making the approach flexible and scalable. Experiments conducted on shape classification and segmentation tasks demonstrate that DD3D consistently outperforms existing baselines. The method proves effective, scalable, and memory-efficient, significantly advancing dataset distillation for 3D data.

## update after rebuttal
The authors tried to address my concerns, and I encourage the authors to fulfill their commitment by including the promised updates in the final manuscript. My final rating is "Weak Accept" (leaning towards accept).

**Claims And Evidence:**

The authors prove mathematically that random rotations weaken the principal components of real data, negatively impacting distillation performance. This strongly motivates the inclusion of a rotation-aware component ("rotator") in their proposed method.

**Essential References Not Discussed:**

There are several references regarding rotation-equivariant and invariant features that are worth discussing:

[1] Zhang et al., RISurConv: Rotation Invariant Surface Attention-Augmented Convolutions for 3D Point Cloud Classification and Segmentation, ECCV’24

[2] Hao et al., RIGA: Rotation-Invariant and Globally-Aware Descriptors for Point Cloud Registration, TPAMI’24

[3] Melnyk et al., TetraSphere: A Neural Descriptor for O (3)-Invariant Point Cloud Analysis, CVPR’24.

**Experimental Designs Or Analyses:**

The approach demonstrates relatively limited performance on fine-grained tasks such as shape classification and part segmentation. This highlights potential limitations in its ability to capture detailed geometric features. Furthermore, the proposed method focuses on shape-level tasks and does not easily extend to large-scale scene-level applications.

Moreover, while the paper compares DD3D with various distillation methods, its claims could be substantially strengthened by including comprehensive benchmarks against state-of-the-art rotation-invariant methods as I will mention below.

**Methods And Evaluation Criteria:**

The authors provide thorough theoretical analyses, formally demonstrating that matching rotation-invariant features is essential for successful 3D point cloud distillation.

**Other Comments Or Suggestions:**

N/A

**Other Strengths And Weaknesses:**

N/A

**Questions For Authors:**

It is good to show experiments with large-scale scene and also compare with recent rotation-invariant methods as I mentioned above.

**Relation To Broader Scientific Literature:**

The paper can promote research on dataset distillation for point clouds, based on 2D image studies.

**Theoretical Claims:**

The paper provides a robust mathematical foundation, clearly explaining why the proposed dataset distillation (DD3D) approach works effectively.

---

> ### Author Rebuttal · Authors · 2025-04-01
>
> We appreciate your detailed review and the recognition of our contributions.
>
> &nbsp;
> ### ``Experimental Designs Or Analyses``
>
> > **Q1: The approach demonstrates relatively limited performance on fine-grained tasks.**
>
> A1: We propose two strategies to further improve the performance of DD3D.
> 1. Improving the number of CPC. We report the performance of DD3D with CPC=100, and find that it can achieve comparable performance with the full dataset.
>
> | ModelNet40 | CPC=50 | CPC=100 | | ScanObject | CPC=50 | CPC=100 |
> | :--: | :--: | :--: | :--: | :--: | :--: | :--: |
> | GM | 81.74 | 84.17 | | GM | 57.52 | 62.82 |
> | DD3D | 83.91 | **86.68** | | DD3D | 61.96 | **65.51** |
> | Full | 88.05 | 88.05 | | Full | 66.96 | 66.96 |
>
> 2. Combined with knowledge distillation (KD). Recent DD methods suggest that leveraging KD can significantly improve the performance of DD. We can also adopt this strategy to achieve lossless performance.
>
> | ModelNet40 | CPC=50 | CPC=100 | | ScanObject | CPC=50 | CPC=100 |
> | :--: | :--: | :--: | :--: | :--: | :--: | :--: |
> |DD3D | 83.91 | 86.68 | |DD3D | 61.96 | 65.51 |
> |DD3D+KD | 86.55 | **88.75** | |DD3D+KD | 65.06 | **66.84** |
> | Full | 88.05 | 88.05 | | Full | 66.96 | 66.96 |
>
> > **Q2: This highlights potential limitations in its ability to capture detailed geometric features.**
>
> A2: The goal of DD is to make the synthetic data informative, making the synthetic samples deviate from the distribution of real data. As a result, the synthetic dataset may have some artifacts and overlook some detailed geometries. **The same results can also be observed in vision dataset distillation.**
>
> Notably, ``Figure 4`` (lines 385-395) illustrates that synthetic data generated by GM has much noise and isolated points, making the data unrealistic. On the other hand, synthetic data generated by DD3D is coherent and captures the global geometric shapes.
>
> > **Q3: The proposed method focuses on shape-level tasks and does not easily extend to large-scale scene-level applications.**
>
> A3: We conduct a semantic segmentation experiment on the S3DIS dataset, which contains more than 80M points in the training data and has more noise than shape-level datasets.
>
> We use Area 5 as the test set and other areas as the training set, and report the overall accuracy (OA) and instance mean IoU of different methods. We can observe that DD3D consistently outperforms GM by a large margin, demonstrating its effectiveness in large-scale scene-level tasks.
> | S3DIS (OA / mIoU) | GM | DD3D | Full |
> | :-- | :--: | :--: | :--: |
> | CPC=1 (0.06%) | 53.09% / 0.3674 | **57.45% / 0.4043** | 73.78% / 0.5786 |
> | CPC=10 (0.6%) | 61.72% / 0.4438 | **64.05% / 0.4624** | 73.78% / 0.5786 |
>
> > **Q4: Including comprehensive benchmarks against state-of-the-art rotation-invariant methods.**
>
> A4: We compare DD3D with some SOTA rotation-invariant methods, including RISurConv, TetraSphere, and SGMNet (suggested by Reviewer gVCR). The results are shown below.
> | ModelNet40 | CPC=10 | CPC=50 | Full |
> | :-- | :--: | :--: | :--: |
> | DD3D | 58.14 | **71.27** | 80.45 |
> | RISurConv | 55.81 | OOM | **95.60** |
> | TetraSphere | 54.72 | OOM | 90.50 |
> | SGMNet | **58.33** | 70.48 | 80.77 |
>
> We have the following observations:
> 1. Both RISurConv and TetraSphere perform well in the full dataset training. However, they are not suitable for the DD task, as they all need to build a k-NN graph to learn rotation-invariant representations. During distillation, the synthetic point clouds are dynamically optimized, resulting in different nearest neighbors. As a result, we need to **re-calculate the k-NN graph** in each iteration, which significantly increases the time and space overhead.
> 2. DD3D and SGMNet are model-agnostic, which can be easily combined with different point cloud models. Here we use the 3-layer PointNet. Their results are better than RISurConv and TetraSphere, indicating that complex models do not always lead to better distillation performance. In contrast, simple models may be more suitable for DD.
>
> &nbsp;
> ### ``Supplementary Material``
>
> > **Q5: The Appendix B is incomplete with only Rotator, there is no Generator. Also, the Algorithm 2 is different with the provided source code. In addition, the provided source code is incomplete too, without instructions to reproduce.**
>
> A5: We will revise our code to make it more complete and easier to read. Specifically,
> - Add a PyTorch-style pseudo algorithm of the Generator.
> - The implementation of Algorithm 2 is rooted in ``CINR.py/InvariantWrapper``. We also provide a simpler vision of the rotator, which removes the hidden dimension of SIREN to reduce the model complexity.
> - Add a Readme file to illustrate the reproduction process.
>
> &nbsp;
> ### ``Essential References Not Discussed``
>
> > **Q6: There are several references regarding rotation-equivariant and invariant features that are worth discussing.**
>
> A6: Thanks for pointing out these important related works. We will cite and discuss them in the revision.

---

> > ### Comment · Reviewer_vcTR · 2025-04-01
> >
> > Thank you the authors for your response, which mostly addressed my concerns.
> >
> > Could you please provide the time and space overhead associated with recalculating the k-NN graph in comparison to rotation-invariant methods? From my experience, the current C++/CUDA implementation of k-NN for Python is very fast, so I doubt it significantly impacts the overall pipeline.

---

> > > ### Author Response · Authors · 2025-04-02
> > >
> > > Thanks for the quick response.
> > >
> > > To provide a detailed comparison, we use ``line_profiler`` to track the time costs in each line.
> > >
> > > We set the ``outer_loop=20`` and ``inner_loop=10`` and report the time consumption of one iteration for gradient calculating, gradient matching, and backward, which dominate the total time costs (>80%).
> > >
> > > | ModelNet40 (CPC=10) | Gradient Time (s) | Matching Time (s)  | Backward Time (s) | GPU Memory (MB) |
> > > | :-- | :--: | :--: | :--: | :--: |
> > > | DD3D | 2.21  | 0.94  | 6.28 | 11320 |
> > > | RISurConv | 43.49  | 9.38 | 57.82 | 24130 |
> > > | TetraSphere | 6.39 | 23.76 | 51.65 | 22504 |
> > >
> > > Notably, this time consumption is only the result of one iteration, and we may need hundreds of iterations in DD. Therefore, the time and space overhead of RISurConv and TetraSphere is larger than that of DD3D.

---

### Decision · Program_Chairs · 2025-05-01

**Decision:**

Accept (poster)

**Comment:**

This paper received positive reviews from all the reviewers. Please include all the promised updates in the final version.